# Mechanisms of extracellular electron transfer in anaerobic methanotrophic archaea

Heleen T. Ouboter[1], Rob Mesman [1], Tom Sleutels [2,3], Jelle Postma[4], Martijn Wissink [1], Mike S. M. Jetten[1], Annemiek Ter Heijne[5], Tom Berben [1] & Cornelia U. Welte [1] ✉

Anaerobic methanotrophic (ANME) archaea are environmentally important, uncultivated microorganisms that oxidize the potent greenhouse gas methane. During methane oxidation, ANME archaea engage in extracellular electron transfer (EET) with other microbes, metal oxides, and electrodes through unclear mechanisms. Here, we cultivate ANME-2d archaea ('Ca. Methanoperedens') in bioelectrochemical systems and observe strong methane-dependent current (91–93% of total current) associated with high enrichment of 'Ca. Methanoperedens' on the anode (up to 82% of the community), as determined by metagenomics and transmission electron microscopy. Electrochemical and metatranscriptomic analyses suggest that the EET mechanism is similar at various electrode potentials, with the possible involvement of an uncharacterized short-range electron transport protein complex and OmcZ nanowires.

The anaerobic oxidation of methane is an important microbial process that regulates the release of methane, a potent greenhouse gas, into the atmosphere. This process is performed by anaerobic methanotrophic ANME-1, −2abc and −3 in marine sediments[1–5], and by ANME-2d in freshwater sediments[6–10]. Our knowledge of the physiology of ANME is heavily reliant on -omics techniques as none of the ANME have been cultured independently of other microorganisms. Marine ANME are dependent on sulfate-reducing bacteria as an electron sink[11–13]. Freshwater ANME can couple methane oxidation to the reduction of nitrate to nitrite, a toxic byproduct. ANME-2d are therefore often found in consortia with nitrite-scavenging bacteria[6,7,14]. Alternatively, ANME can use insoluble metal oxides as electron acceptors, which likely makes these archaea an important methane sink in iron-rich sediments[15–19].ANME-2d, which belong to the *Methanoperedenaceae* family, can use humic substances, manganese oxides, iron oxides, selenate, arsenate, and chromate as the electron acceptor for the conversion of methane[20–27]. For electroactive microorganisms the challenge is to adapt their protein machinery to tune into the different surface redox potentials of insoluble electron acceptors[28] such as metal oxides, which are omnipresent in sediments. Although the theoretical free energy gain increases the more positive the electron acceptor redox potential couple is, the enzymatic machinery of the microorganisms eventually determines whether this increased theoretical yield can be capitalized on[28]. Previous research has demonstrated that the extracellular electron transfer (EET) machineries of the model electroactive microorganisms *Geobacter* and *Shewanella* involve various extracellular electron transfer proteins, many of which contain multi-heme *c*-type cytochromes (MHCs) that form electron conduits. *Geobacter* encodes triheme periplasmic-type cytochromes (Ppc)[29,30], pili[31] and several outer-membrane cytochromes (OMCs) as part of its EET

[1]Department of Microbiology, Radboud Institute for Biological and Environmental Sciences, Radboud University, Heyendaalseweg 135, 6525AJ Nijmegen, The Netherlands. [2]Wetsus, European Centre of Excellence for Sustainable Water Technology, Oostergoweg 9, 8911 MA Leeuwarden, The Netherlands. [3]Faculty of Science and Engineering, University of Groningen, Nijenborgh 4, 9747 AG Groningen, The Netherlands. [4]Department of General Instrumentation, Radboud University, Heyendaalseweg 135, 6525AJ Nijmegen, The Netherlands. [5]Environmental Technology, Wageningen University & Research, Bornse Weilanden 9, 6708 WG Wageningen, The Netherlands. ✉e-mail: c.welte@science.ru.nl

machinery[29,32]. Similar to these electroactive bacteria, '*Ca*. Methanoperedens' species have a large MHC repertoire, some of which are expressed during growth with iron oxides[20], manganese oxides[24], or nitrate as the electron acceptor[33]. However, the mechanism of extracellular electron transfer in ANME archaea remains largely unresolved. While some MHCs have been implicated in EET, comparative physiology studies of ANME archaea are scarce due to the limitations imposed by their slow growth rates in complex communities, and cryptic cultivation requirements. These problems are compounded by the presence of bacteria known to perform extracellular electron transfer, such as *Geobacter*, in *Methanoperedenaceae*-dominated enrichment cultures. It has been suggested that soluble intermediates produced by ANME archaea (i.e. acetate) are utilized by other electroactive bacteria, which then perform the ultimate reduction of the extracellular electron acceptor[34–36].

In this study, we aimed to investigate extracellular electron transfer by '*Ca*. Methanoperedens' ANMEs using bioelectrochemical systems at various anode potentials compared to nitrate-grown conditions to perform comparative physiology experiments. Here we present the resulting bioelectrochemical data, visualization of the anode biofilm using fluorescence and electron microscopy, and metagenomics and metatranscriptomics analyses of the composition and activity of said biofilm. We reached a strong methane-dependent current that was not dependent on the anode potential. Analysis of gene expression showed upregulation of two gene clusters possibly involved in EET. At the same time, we obtained highly enriched microbial communities dominated by '*Ca*. Methanoperedens' that might open doors to future axenic ANME cultures.

## Results

In total, three experiments were performed to investigate the mechanism of extracellular electron transfer (EET) by '*Ca*. Methanoperedens' to a gold anodic electrode mesh by several tests and sampling for metagenomic and metatranscriptomic analysis and microscopy imaging (Fig. 1). In experiment 1 (Fig. S1), there was a start-up phase of ~1 week where four parallel systems were run at 0 mV vs SHE to develop a comparable biofilm and a subsequent phase of two more weeks where the potential was kept at 0 mV in BES1 or switched to 200 mV (BES2), 400 mV (BES3), or 600 mV (BES4). In experiment 2 (Fig. S2), the microbial community was incubated for 6.5 weeks at 0 mV vs SHE in a BES. In experiment 3 (Fig. S3), the microbial community was incubated for 9 weeks at 0 mV vs SHE in a BES, with a 3 week 'famine' phase in which methane oxidation was disabled by setting the electrode potential to −400 mV vs SHE (Fig. S4).

### Methane-dependent current production remains unchanged at varying anode potentials

To investigate the mechanism of EET by '*Ca*. Methanoperedens' and the influence of the electric potential on the process, we conducted bioelectrochemical experiments at four different anode potentials: 0 mV, 200 mV, 400 mV, and 600 mV vs SHE (Fig. S1). In each BES, the initial biofilm was established under near-identical conditions at a potential of 0 mV vs SHE, after which the potential was changed, and current production and the microbial community were monitored (experiment 1). No clear effect of the poised potential on the amount of produced current was observed (Fig. 2A), while a strong methane-dependent current ($59\% \pm 11\%$ out of $30 \pm 8.8$ mA m$^{-2}$) was produced at all potentials (Fig. 2B, Fig. S5). Anaerobic methanotrophic activity was confirmed through the increase of $^{13}CO_2$ over total $CO_2$ after the culture was supplied with $^{13}CH_4$ (Fig. S5). The microbial community consumed on average $71 \pm 0.017$ µmol methane day$^{-1}$ which was accompanied by the production of $3.0 \pm 1.5$ C (Fig. S5). Cyclic voltammetry scans at different anode potentials were nearly identical, suggesting that the same redox centres were operational under the four different poised anode conditions (Fig. 2C). Two midpoint redox potential signals could be identified from the cyclic voltammetry scans by averaging the oxidative and reductive peaks: one at -0.18 V and one at +0.10 V, indicating that at least two distinct redox protein complexes were active in the microbial community colonizing the anode. Especially the more negative redox centre is likely to be involved in the electron transfer from $CH_4$ to the anode, due to its low potential. Polarization curves indicated that the potential at which the microbial community began to produce current was close to the thermodynamic redox potential of the $CH_4/CO_2$ redox couple at −0.249 V (with conditions: pH 7.25 and a gas phase of 86.4% $CH_4$ and 4.55% $CO_2$) indicating that methane was the most likely electron donor to the anode (Fig. 2D). Furthermore, if organic material would have been the electron donor to the detected current production, the observed intercept of the polarization curve with the *x*-axis should have been closer to the acetate/$CO_2$ redox couple (-0.29 V). The shape of the polarization curves was similar, suggesting that the EET mechanism was not influenced by the applied potential (Fig. 2D).

### Long-term incubations in bioelectrochemical systems led to anode biofilms highly enriched in '*Ca*. Methanoperedens'

After having established stable methane-dependent current production at 0 V in experiment 1 (Fig. S1), we prolonged the incubation time, performed additional medium refreshments, and monitored the percentage of methane-dependent current in experiment 2. The relative abundance of '*Ca*. Methanoperedens' was determined at the end of each incubation (experiments 1, 2; Fig. 3), as sampling of the electrode

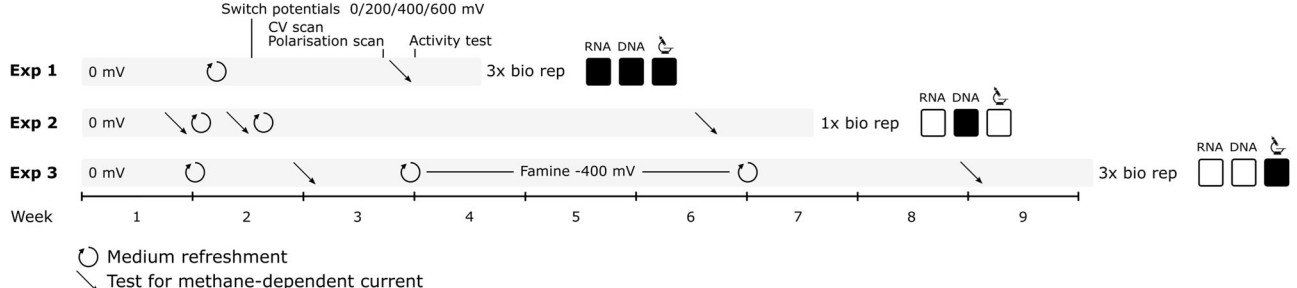

**Fig. 1 | A schematic overview of the three performed experiments.** Events during the experiments such as medium refreshments, test for methane-dependent current, cyclic voltammetry (CV) and polarisation scans, activity test, and sampling for metagenomics and metatranscriptomics analysis as well as microscopy visualization are indicated. All systems in Experiment 1 were run in biological triplicate; RNA samples were collected at 0 mV in triplicate (2 biological, 1 technical replicate; $n = 2$), at 200 mV in triplicate (3 biological replicates, $n = 3$), at 400 mV in

quadruplicate (3 biological replicates, 1 technical replicate, $n = 3$), at 600 mV in quadruplicate (3 biological replicates, 1 technical replicate, $n = 3$), and under nitrate growth in triplicate (3 biological replicates, $n = 3$). The differences in RNA samples for transcriptomics were a result of low amounts of biomass obtained and challenges associated with RNA extraction in archaea. DNA extraction for metagenomics was performed from one bioanode per experimental condition in experiment 1, and once during experiment 2.

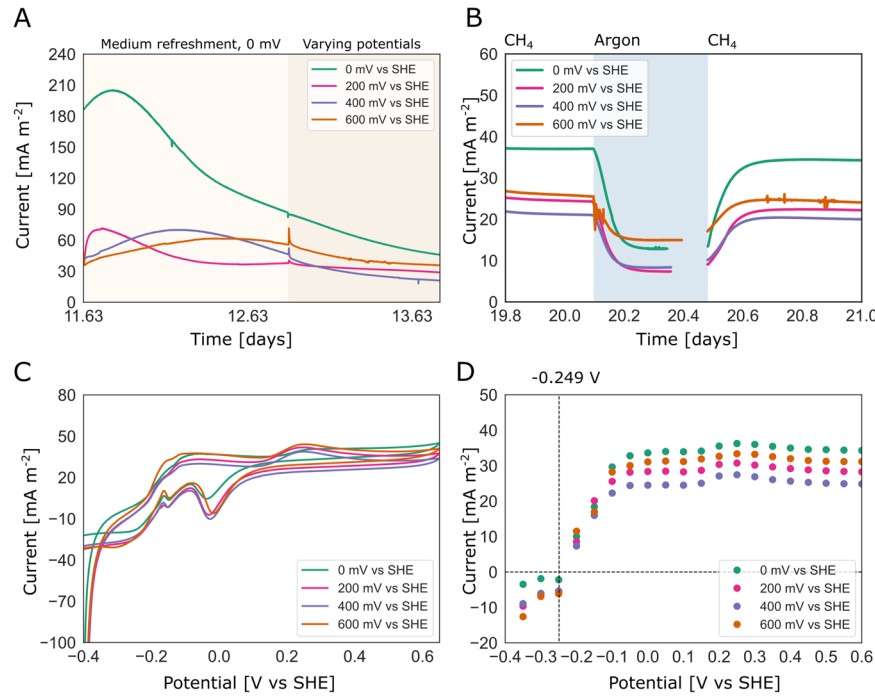

**Fig. 2 | Bioelectrochemical data obtained during experiment 1, including measurements taken under various potentials.** The figures show the average of three biological replicates ($n = 3$). **A** The current production after the medium was replaced and after all systems were initially operated at 0 mV vs SHE, followed by a switch to different potentials (0 mV, 200 mV, 400 mV, or 600 mV vs SHE). **B** The replacement of methane by argon to test for methane-dependent current. **C** Cyclic voltammetry scans. **D** Polarisation scans with the redox potential of the $CH_4/CO_2$ redox couple indicated by the dashed line at −0.249 V. Green, 0 V vs. SHE condition; pink, 200 mV vs. SHE condition; purple, 400 mV vs. SHE condition; orange, 600 mV vs. SHE condition. Source data are provided as a Source Data file.

biofilm requires sacrificing the experiment. We conducted separate experiments with different incubation times and medium refreshments (experiment 1 and Fig. S1; Experiment 2 and Fig. S2). Our results showed that a higher number of medium refreshments and a longer incubation time were associated with higher percentages (up to 91%) of methane-dependent current and higher relative abundance (up to 82%) of '*Ca.* Methanoperedens' at the anode determined by metagenomics (Fig. 3 and Fig. S6).

In experiment 3, during three out of 9 weeks we applied a potential of −400 mV vs SHE (Fig. S3, Fig. S4) to disable methane oxidation at the bioanode. By applying a potential that is lower than −0.24 mV, which is the potential of redox couple $CH_4/CO_2$, we tested whether anaerobic methane oxidation can be reversed by '*Ca.* Methanoperedens'. This period turned out to be a starvation period as we observed that the culture could only produce little methane ($0.22 \pm 0.029$ µmol methane day$^{-1}$) and a stable current of −0.08 mA m$^{-2}$ which shows that they can only reach at best a methane production rate that is 0.31% of the methane oxidation rate. This might indicate that '*Ca.* Methanoperedens' can reverse their metabolism at extremely low rates which are probably insignificant in their natural habitat, yet more experiments are needed to substantiate the observation that a low amount of cathodic electron uptake results in methane production by '*Ca.* Methanoperedens'. Specifically, our metagenomics data indicated the absence of methanogens in the microbial community yet more detailed analyses are necessary to confirm this observation. Nonetheless, the microbes remained active even after the starvation period with 41 mA m$^{-2}$ of stable current before starvation vs. 61 mA m$^{-2}$ after starvation (Fig. S3). After starvation, when the potential was returned to 0 mV vs SHE, 93% methane-dependent current was produced (Fig. S6E, Fig. S3).

## Imaging revealed spatial organization of the biofilm microbial community
Confocal laser scanning microscopy combined with fluorescence in-situ hybridization (FISH), scanning electron microscopy (SEM), and

transmission electron microscopy (TEM) were employed for samples from experiment 3 to visualize the biofilm on the anode (Fig. 4). The signals for '*Ca.* Methanoperedens' and archaea fully overlapped indicating that no other archaea than '*Ca.* Methanoperedens' were present (Fig. S7), congruent with the meta-omics data (Fig. 5). The confocal laser scanning micrographs revealed a large number of '*Ca.* Methanoperedens' cell aggregates (Fig. 4A, D); the fluorescent signals of '*Ca.* Methanoperedens' aggregates, together with the laser reflection used to visualize gold, were both observable in a confocal volume of 500 nm, governed by the system's optical resolution determined by the used 63x/1.40 lens. This is evidence that '*Ca.* Methanoperedens' are present further than 500 nm away from the anode gold mesh (Fig. 4A) and that they were also present within 500 nm of the gold mesh (Fig. 4D). Since *Geobacter* sp. is a well-known electro-active bacterium hypothesized to be involved in EET by '*Ca.* Methanoperedens' and part of the microbial community in the inoculum[33], we targeted this bacterium specifically. Bacteria were observed near the mesh, and a large proportion of these bacteria hybridized with the *Geobacter* probe (Fig. 4H, I). Scanning electron microscopy revealed granules with the typical cell morphology of '*Ca.* Methanoperedens', which covered a large portion of the electrode (Fig. 4J). Correspondingly, transmission electron microscopy showed aggregated sarcina-shaped clusters, with intact cells containing storage compounds that were represented as white dots within the granule (Fig. 4K). The clusters were located at varying distances from the gold electrode mesh with a minimum of 4 µm (Fig. 4K), corresponding with the FISH observations, and a maximum of 45 µm (Fig. S8). Furthermore, the transmission electron micrographs showed bacterial cells in the proximity of the gold mesh few of which showed intracellular structures (Fig. 4L).

## Antibiotics targeting bacteria do not affect current production
To unravel the relative contribution of bacteria and archaea to the current production, we incubated the anode biofilms with antibiotics, consisting of vancomycin, streptomycin, ampicillin and kanamycin,

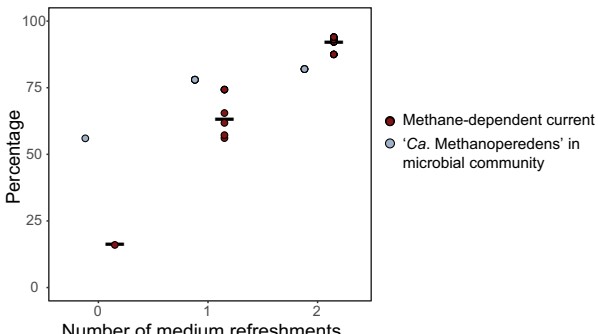

**Fig. 3 | Influence of the number of medium refreshments and incubation time on the percentage of methane-dependent current and relative abundance of 'Ca. Methanoperedens'.** The relative abundance [in %] of 'Ca. Methanoperedens' in the microbial community is depicted as blue circles whereas red circles signify the percentage of methane-dependent current, both plotted relative to the number of medium refreshments. Metagenomics data for abundance estimations were collected in one biological replicate ($n = 1$) per condition. Methane-dependent current data were collected in one biological replicate ($n = 1$) for the condition "without medium replacement", in five biological replicates ($n = 5$) for one medium refreshment, and in four biological replicates ($n = 4$) for two medium refreshments. Mean values of methane-dependent current are indicated as horizontal lines. Source data are provided as a Source Data file.

targeting only bacteria and not archaea. The current did not change after the addition of antibiotics, with a small increase of current observed right after the antibiotic addition (Fig. S9, Fig. S10, Fig. S11). After several days of incubation, the current did not drop below the current density measured before the addition of antibiotics, and the methane-dependency of the current increased by 2%, suggesting that current-producing bacteria were either not affected by the addition of antibiotics or were not responsible for current generation. To further test the hypothesis that 'Ca. Methanoperedens' is the principal driver of current generation we added 2-bromoethanesulfonate, which inhibits the key methanotrophy enzyme methyl-coenzyme M reductase (MCR), and puromycin, an antibiotic that affects the RNA translation with a more pronounced effect on archaea compared to bacteria. The addition of 20 mM 2-bromoethanesulfonate resulted in an immediate and strong reduction in current by 89% (Fig. S11), suggesting that a small proportion of the current may be generated by the conversion of storage polymers, such as polyhydroxyalkanoate (PHA), which were visible in the TEM micrographs (Fig. 4K), a process that is independent of MCR, or by bacteria associated to the electrode. Upon the addition of puromycin, the current showed a gradual decrease, indicating that the antibiotics successfully penetrated the biofilm and affected the archaea (Fig. S10); at the same time, it establishes that the remaining current was due to residual activity of enzymes in puromycin-affected archaeal cells, unlike what was seen in the treatment with bacterial-targeting antibiotics, where no gradual decrease of current density was observed (Fig. S10 and Fig. S11).

### Global metagenome and metatranscriptome analysis

Metagenomics and metatranscriptomics analyses were conducted to further determine the composition of the microbial community and the activity of its members (Fig. 5, Fig. S12). The relative abundance of 'Ca. Methanoperedens' was 54% under nitrate reducing conditions and reached a maximum of 78% in the anode biofilm after 3 weeks of incubation (Fig. 5) and 82% after 9 weeks of incubation (Fig. 3 and Fig. S3). The transcriptional activity was measured as the amount of RNA-seq reads mapping to a specific genome relative to the number of reads mapping to the total metagenome. 'Ca. Methanoperedens' represented ~20% of the total activity under nitrate reducing conditions, whereas it represented 45–64% of the total activity in the BES when the electrode was used as an electron acceptor (Fig. 5B, Fig. S12).

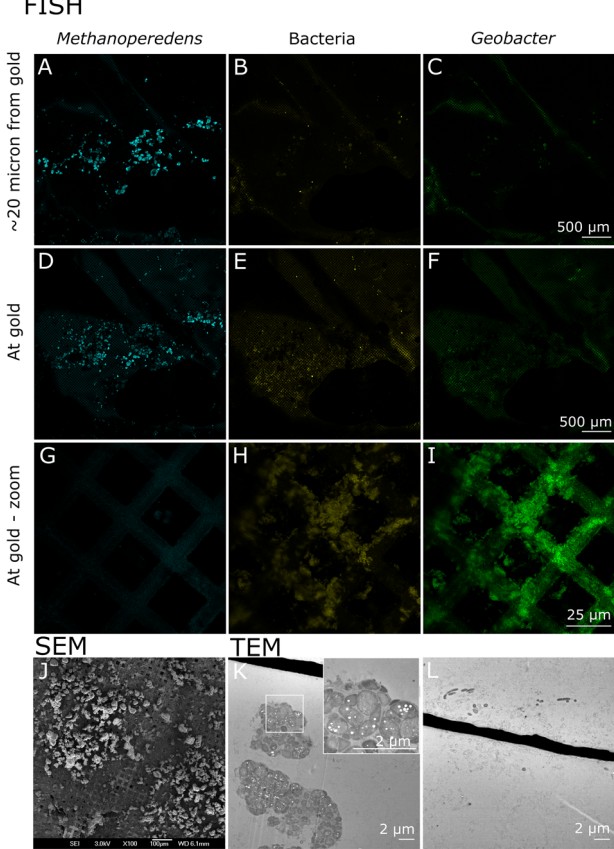

**Fig. 4 | Visualization of the biofilm using different microscopy techniques.** **A–I** Confocal laser scanning microscopy combined with fluorescence in situ hybridization (FISH) to target 'Ca. Methanoperedens' using a FLUOS probe (cyan), bacteria using a Cy5 probe (yellow), and Geobacter sp. using a Cy3 probe (green), **J**, scanning electron microscopy (SEM) showing large clusters resembling 'Ca. Methanoperedens' cells further away from the electrode mesh and a dense biofilm of rod-shaped bacteria closer to the gold mesh, **K**, **L** transmission electron microscopy (TEM) indicating 'Ca. Methanoperedens' cells (**K**) further away from the gold mesh and rod-shaped bacterial cells (**L**) closer to the gold mesh. All the samples shown in this figure were obtained from experiment 3. Similar results were observed in three biological replicates ($n = 3$) for **A–H** and two biological replicates ($n = 2$) for **C**, **F**, **I**; in biological duplicate ($n = 2$) for scanning electron microscopy (**J**) and biological duplicate ($n = 2$) for transmission electron microscopy (**K**, **L**).

Several proteobacteria were found to be active in the nitrate condition which were not active in the BES. Geobacteraceae and Ignavibacteriaceae, despite their low relative abundance, were found to be active under all conditions. These bacteria are known for their electroactive properties and were probably using organic matter derived from decaying biomass or excreted products as their electron donor. The completeness of the 'Ca. Methanoperedens', Geobacter sp. and Ignavibacteriaceae MAGs were high with 99.3%, 99.4% and 97.2% respectively.

### Transcriptomics analyses reveal two differentially expressed gene clusters most likely involved in extracellular electron transfer

The 'Ca. Methanoperedens' MAG contains genes for the complete reverse methanogenesis pathway in addition to several copies detected in the unbinned contigs of the metagenome. Although several strains of 'Ca. Methanoperedens' were present in the BES, only one was highly abundant and transcriptionally active (Supplementary Data 1). The genes within the MAG encoding proteins of the reverse methanogenesis were upregulated in the electrode condition compared to the nitrate-reducing condition indicating the high activity of 'Ca.

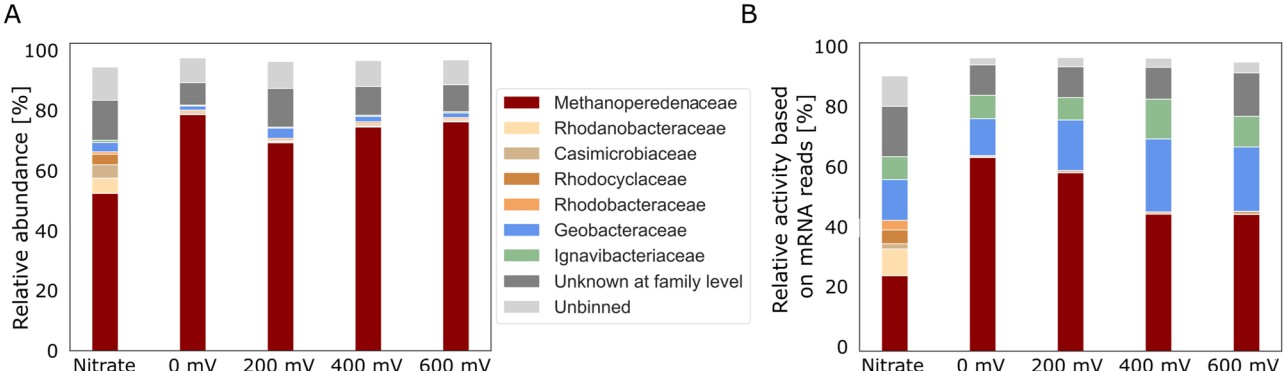

**Fig. 5 | Metagenomic and metatranscriptomic analysis of the microbial community.** Relative abundance and relative activity at the family level determined by mapping (**A**) metagenome reads and (**B**) RNA-seq reads to the metagenome-assembled genomes (MAGs). The families *Methanoperedenaceae* and *Geobacteraceae* both comprise a single MAG that was identified as '*Ca*. Methanoperedens' and *Geobacter* sp., respectively, based on the classification by the Genome Taxonomy Database (GTDB). Only taxa with a relative abundance or activity of >2.5% are displayed. RNA samples were collected at 0 mV in triplicate (2 biological, 1 technical replicate; *n* = 2), at 200 mV in triplicate (3 biological replicates, *n* = 3), at 400 mV in quadruplicate (3 biological replicates, 1 technical replicate, *n* = 3), at 600 mV in quadruplicate (3 biological replicates, 1 technical replicate, *n* = 3), and under nitrate growth in triplicate (3 biological replicates, *n* = 3). The differences in RNA samples for transcriptomics were a result of low amounts of biomass obtained and challenges associated with RNA extraction in archaea. DNA extraction for metagenomics was performed from one bioanode per experimental condition in experiment 1, and once during experiment 2. Source data are provided as a Source Data file.

Methanoperedens'. Using metatranscriptomics, we identified two strongly upregulated gene clusters in this MAG that encoded multiple multi-heme *c*-type cytochromes (MHCs). The first gene cluster contained two 7-heme binding MHCs (Metp_01715, Metp_01720) and was co-located with genes encoding an NrfD-like *b*-type cytochrome-containing transmembrane protein (Metp_01713), a protein predicted to harbour four [4Fe-4S] clusters (Metp_01714), an Fdh-like membrane-integral *b*-type cytochrome (Metp_01719), and unknown proteins (Metp_01716-18, Metp_01721-22) (Fig. 6). While some genes in this cluster were expressed under both electrode and nitrate conditions, others were upregulated in the electrode condition (Supplementary Data 1, Fig. 6). The second gene cluster contained genes that encode proteins homologous to those found in *Geobacter sulfurreducens* related to the formation of OmcZ nanowires[37]: Metp_02468 encodes a protein that is homologous to OmcZ from *Geobacter sulfurreducens* and Metp_02469 encodes a protein that is homologous to a serine protease OzpA from *G. sulfurreducens* that is required for the assembly and folding of the OmcZ extracellular nanowire. Guo et al.[37]. have also suggested structural homology through comparison of the '*Ca*. Methanoperedens' OmcZ AlphaFold model to the *Geobacter sulfurreducens* OmcZ nanowire structure determined by cryo-electron microscopy.

All genes in this cluster were strongly expressed (Supplementary Data 1) and upregulated in the electrode condition, with log2-fold changes of at least 3.6 (Fig. 6). A schematic overview of the proposed metabolism of '*Ca*. Methanoperedens' and its postulated EET mechanism is depicted in Fig. 7.

We also mined the '*Ca*. Methanoperedens' MAG for genes encoding multi-heme *c*-type cytochromes and found fifteen out of a total of thirty-three genes encoding MHCs upregulated (log2 fold change > 2) at the electrode compared to the nitrate condition (Supplementary Data 1). One of these genes encoded an MHC with an S-layer domain suggesting its involvement in the electron transfer through the outermost cellular layer. Additionally, all three genes encoding major subunit flagellin (*flaB*) that were found in the *Methanoperedens* MAG were upregulated in the electrode condition compared to nitrate growth. *Geobacter* and *Ignavibacteriaceae* were both active at the electrode (Fig. 6): *Geobacter* expressed genes related to their EET mechanism together with genes related to acetate metabolism (Supplementary Data 1), but none of these genes was significantly upregulated (log2 fold change > 2) at the electrode compared to the nitrate condition. For

*Ignavibacteriaceae* the EET mechanism is not known but we found upregulated genes encoding MHCs (Fig. 6, showing *Ignavibacteriaceae* MHCs > 30 normalized counts). Genes related to their acetate metabolism were expressed but not upregulated (Supplementary Data 1).

The metatranscriptome analysis indicated that the expression patterns observed for the four different potentials were similar, as previously observed in the bioelectrochemical data (Fig. 2, Fig. 6).

## Discussion

In this work, we demonstrated the ability of '*Ca*. Methanoperedens' to perform extracellular electron transfer (EET) to an electrode at potentials ranging from 0 mV to 600 mV vs standard hydrogen electrode (SHE). We discovered that during this process, '*Ca*. Methanoperedens' expresses a gene cluster encoding for proteins with structural similarities to the OmcZ nanowire-related proteins found in *Geobacter sulfurreducens* used for long-range (100 μm) electron transfer with the highest expression at 0 mV[37]. Unlike the OmcZ nanowire found in *Geobacter*, '*Ca*. Methanoperedens' encodes two additional multiheme *c*-type cytochromes (MHCs) with 37 and 15 heme-binding sites, respectively, in the same gene cluster. We identified that this gene cluster was highly upregulated (log2-fold change > 3.5) under electrogenic growth conditions compared to when nitrate was used as an electron acceptor (Supplementary Data 1). Surprisingly, varying the potential from 0 to 600 mV vs. SHE did not influence the EET mechanism of '*Ca*. Methanoperedens', in contrast to what was suggested by Zhang et al.[38]. One reason for this could be the sub-optimal growth of '*Ca*. Methanoperedens' in their study, evidenced by the 6-100-fold lower current production by their anodic biofilm and lower relative abundance in the electrode biofilm (ca. 7% versus >80% in our study). The unchanged EET mechanism in our study was evidenced by the presence of unchanged redox centres observed during the cyclic voltammogram scans across the different tested conditions, the comparable shape of the polarization scans and the absence of potential-induced changes in current production and gene expression patterns (Fig. 2, Fig. 6, Supplementary Data 1). This suggests that '*Ca*. Methanoperedens' may be using diverse extracellular electron acceptors in the environment without the need to change its expression pattern or EET mechanism, unlike what was reported for *Geobacter* which can fine tune its EET mechanism in response to a presented redox potential[39–41]. *Geobacter's* requirement for such adaptation may be attributed to the significant number of

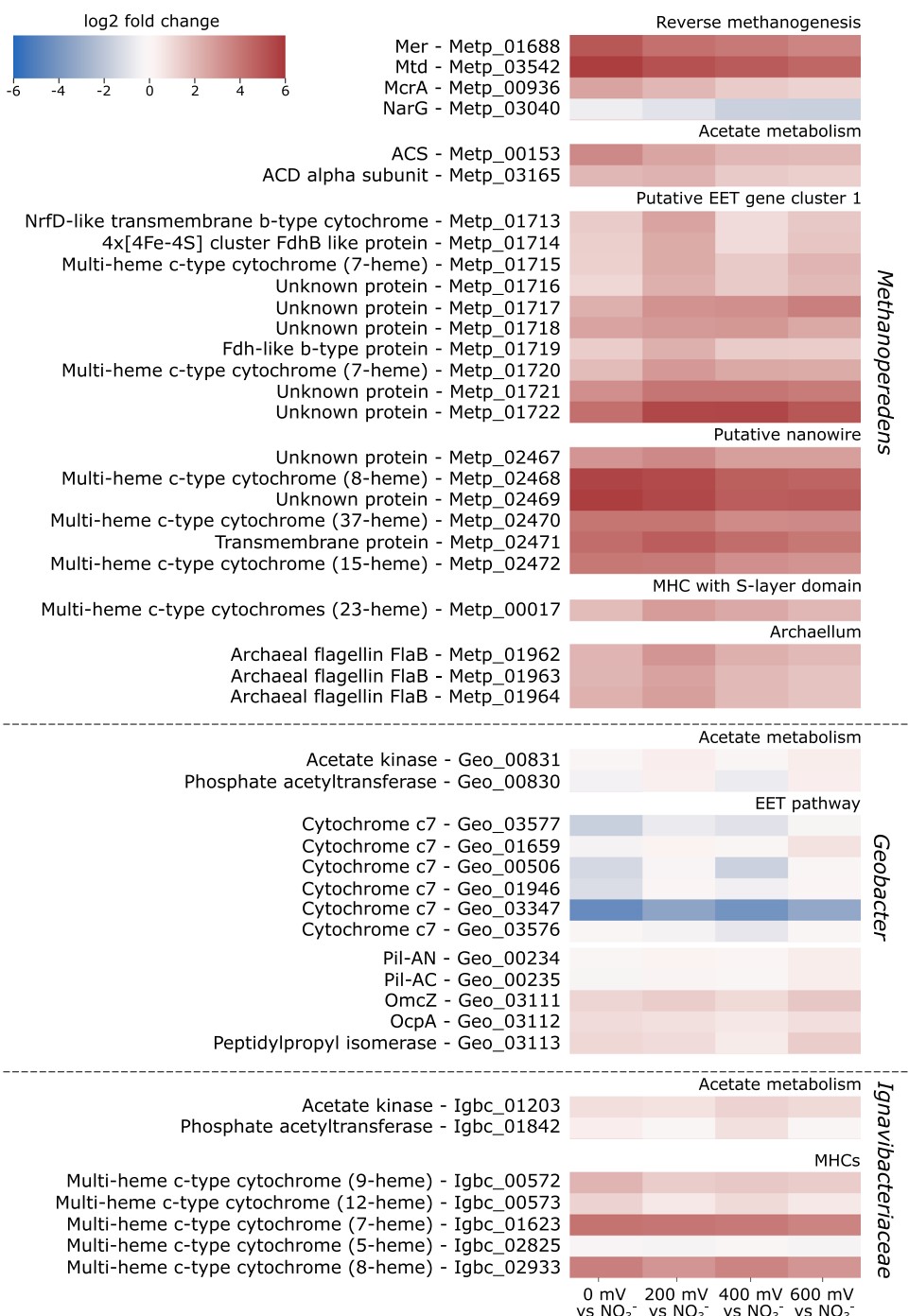

**Fig. 6 | Differential gene expression analysis comparing the electrode condition with potentials 0 mV, 200 mV, 400 mV or 600 mV to the nitrate condition.** The figure displays two gene clusters from 'Ca. Methanoperedens' that are potentially involved in EET, along with genes involved in the EET mechanism of *Geobacter* sp. Additionally, it shows MHCs from *Ignavibacteriaceae* with at least thirty normalized counts in the electrode condition, as the EET mechanism of this microorganism is currently unknown. *Geobacter* and *Ignavibacteriaceae* were specifically mentioned as they were found to be active in the study. Furthermore, genes encoding enzymes involved in acetate metabolism from these three microorganisms were added, as they may play a role in the interaction between 'Ca. Methanoperedens' and the two bacteria. Source data are provided as a Source Data file.

heterotrophic competitors it encounters, whereas for 'Ca. Methanoperedens', this number is probably considerably smaller, given its status as chemolithoautotroph.

Although EET by 'Ca. Methanoperedens' has been studied both in bioelectrochemical systems[36,38,42–45] and in relation to extracellular electron acceptors[17,23], in most studies the EET mechanism has not further been unravelled. Three exceptions are studies focusing on the use of ferrihydrite[20], birnessite[24] and a poised electrode at +400 mV by 'Ca. Methanoperedens'[38], which identified several MHCs expressed

during metal-oxide reduction and electrogenic growth. Similar to 'Ca. Methanoperedens manganicus'[24] the *Methanoperedenaceae* member in this study expressed genes encoding a large subunit flagellin (*flaB*) that is part of the archaellum which has been found to be electrically conductive in *Methanospirillum hungatei*[46]. Studies investigating the role of MHCs containing an S-layer domain in extracellular electron transfer have reported varying results depending on the electron acceptor used. Leu et al. have shown the expression of such MHCs during manganese-dependent methanotrophy[24], while no such MHCs

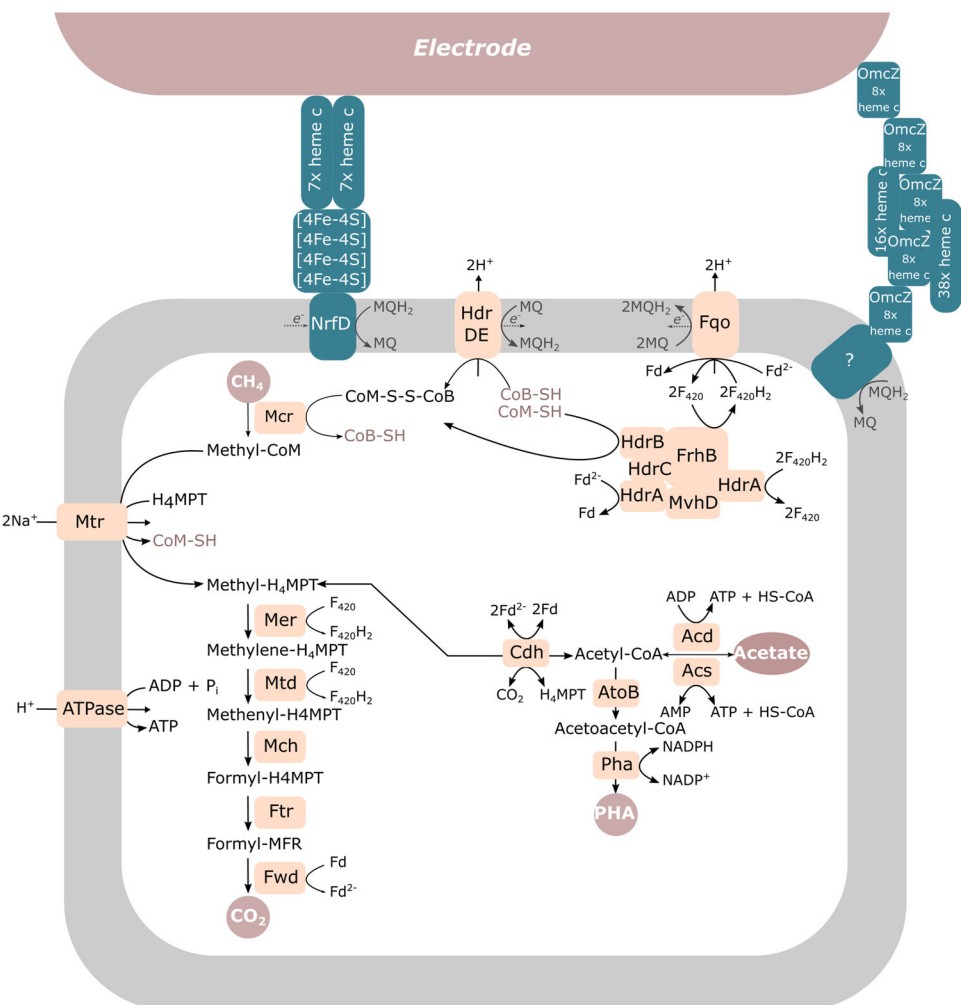

**Fig. 7 | Overview of proposed metabolism from '*Ca.* Methanoperedens' and its putative EET mechanism towards the electrode.**

were expressed during EET to iron minerals or syntrophic growth with sulfate-reducing bacteria[20,47]. We found one MHC with 23 heme-binding motifs in our '*Ca.* Methanoperedens' MAG whose gene also encoded an S-layer domain as described by McGlynn et al. [13]. The gene was differentially expressed during electrogenic growth compared to nitrate-dependent growth. The presence of this MHC with a fused S-layer domain suggests its involvement in electron transfer across the outer layer of the archaeon while using the electrode as electron acceptor.

The expression of a second gene cluster containing genes encoding for MHCs, an [4Fe-4S] cluster protein and *b*-type cytochromes was observed during both electrode utilization and the use of nitrate as an electron acceptor. While it was strongly upregulated under growth on electrodes with the highest expression at +200 mV vs SHE, it had a high baseline expression under nitrate-dependent growth conditions (with an average of 202 normalized counts for the genes in this cluster) compared to genes involved in reverse methanogenesis (*mer*, 39; *mch*, 6; *mtd*, 15). Given these findings, this gene cluster may be involved in a process unrelated to the reduction of solid electron acceptors, such as direct interspecies electron transfer (DIET), like the process observed between *Geobacter* and methanogens[48–51], or short-range electron transfer to extracellular electron acceptors. From our experiments, it is not possible to link the midpoint redox potential of the cyclic voltammetry scan unequivocally to one of the putative EET protein complexes; further experiments are required to do so. Interestingly, Leu et al. identified a similar gene cluster in '*Ca.* Methanoperedens manganicus' that was found to be highly expressed during

growth with birnessite, making its involvement in extracellular electron transfer likely.

The inaccessibility of a pure culture of any of the sub-clades of anaerobic methanotrophic (ANME) archaea has created a significant knowledge gap regarding their physiology. ANME archaea are reliant on syntrophic partners to act as electron sinks[11–13] or to convert nitrite[6,7,14], which is a toxic byproduct of the independent reaction ANME-2d can perform converting methane at the expense of nitrate. In this study, we successfully overcame this obstacle by cultivating ANME-2d on an electrode. We obtained a culture where '*Ca.* Methanoperedens' comprised 82% of the total community, which was achieved in a timeframe of only a couple of weeks. This was accompanied by a produced current that was 93% directly dependent on methane, suggesting that bioelectrochemical systems might be key in further enriching and possibly eventually obtaining axenic ANME cultures. Although *Geobacter* and *Ignavibacteriaceae* were active at the electrode, the lack of effect of bacterial antibiotics streptomycin, vancomycin, ampicillin, and kanamycin on the methane-dependent current and the significant impact of 2-bromoethanesulfonate, an MCR inhibitor, on the current suggests a low contribution of these bacteria to the total current produced; these observations need to be further corroborated in future experiments. Even though *Geobacter* sp. is known to be inhibited by streptomycin and kanamycin[52], the bacterial biofilm growth on the electrode might have prevented the antibiotics from reaching their target thereby not excluding the possibility that the methane-dependent current is produced via *Geobacter* through direct interspecies electron transfer[34–36]. The intact '*Ca.*

Methanoperedens' clusters visualized using various microscopic techniques provide further evidence for 'Ca. Methanoperedens' to be the driving microorganism for the produced current and confirms that 'Ca. Methanoperedens' can perform electrogenic anaerobic oxidation of methane, possibly independently of syntrophic partners, in line with what was suggested by Zhang et al. [38].

In conclusion, we present evidence for current production and extracellular electron transfer by 'Ca. Methanoperedens' during methane oxidation, either through DIET with *Geobacter* sp. or independently, possibly via cytochrome-containing nanowires. These findings have implications for anaerobic methanotrophs in the environment as most forms of methanotrophy rely on extracellular electron transfer; these methanotrophs together represent the major anoxic methane sink on our planet with strong implications for climate change. At the same time, electricity production from methane offers exciting opportunities for future sustainable biotechnological applications.

## Methods

### Cultivation of 'Ca. Methanoperedens' in bioelectrochemical systems

The inoculum was obtained from an enrichment culture dominated by 'Candidatus Methanoperedens' seeded from Vercelli rice fields that was exposed to nitrate limiting conditions and continuously operated since March 2014[7,33]. This culture was grown on electrodes in a two-chamber bioelectrochemical system (BES)[33] (anode: total volume 350 mL, working volume 300 mL; cathode: total volume 310 mL, working volume 280 mL) connected via a RALEX® heterogeneous cation-exchange membrane (CMHPES) (Mega, Prague, Czech Republic). The BES medium consisted of (per L) 0.1 g $CaCl_2 \cdot 2H_2O$, 0.1 g $MgCl_2 \cdot 4H_2O$, 0.05 g $KH_2PO_4$, 0.5 g $NH_4Cl$, 2.38 g HEPES and was autoclaved prior to the addition of 2 mL $L^{-1}$ trace elements[7] and 0.1 mL vitamin solution[7] with final pH of 7.25 and room temperature. After inoculation, 0.3 mL $FeSO_4$ (10 mM) was added to the anaerobic anolyte and the anode chamber was continuously sparged with $CH_4/CO_2$ 95%/5% at 10 mL $min^{-1}$ and $N_2$ at 2.2 mL $min^{-1}$ while the cathode chamber was continuously sparged with $N_2$ at 10 mL $min^{-1}$. The anode electrode (7.7 × 1.9 cm) consisting of gold mesh foil (Precision Eforming, New York, US), the cathode made of stainless-steel mesh (Goodfellow, Huntingdon, UK) and a 3 M Ag/AgCl reference electrode (Prosense, Oosterhout, NL) were connected to a MultiEmStat3 potentiostat (PalmSense, Houten, NL). Anode and cathode were connected via a platinum wire (Goodfellow, Huntingdon, UK). Three experiments were performed (Fig. 1): in experiment 1, the BES was operated in two stages for a total duration of 3 weeks. In stage 1 (7–8 days) the anode was poised at a potential of 0 V vs SHE using a MultiEmStat3 potentiostat (PalmSense, Houten, Netherlands) to develop a biofilm under comparable conditions. At the end of stage 1, the medium containing planktonic cells was replaced by fresh medium and the electrode potential was set at a variety of potentials: 0.0 V, 0.2 V, 0.4 V and 0.6 V vs SHE. Stage II of the experiment was run for 14 d. At the end of the experiment RNA, DNA and microscopy samples were collected anaerobically in an anaerobic hood (97% $N_2$, 3% $H_2$). To compare RNA expression between the microbial community grown at the electrode and with nitrate as electron acceptor, a batch experiment was performed with the medium as mentioned above but with the addition of 3 mM nitrate, without the addition of a poised electrode. This experiment was run for 5 days after which samples were collected anaerobically for DNA and RNA sequencing. In experiment 2, the BES was operated at 0 mV including two medium refreshments with a total duration of 6.5 weeks to investigate the effect of a longer incubation on the community and the obtained current. At the end of the experiment DNA samples were collected anaerobically for metagenomic sequencing. In experiment 3, the BES was operated for 9 weeks at 0 mV including three medium refreshments and a 3 week famine phase at a

potential of -400 mV. At the end of the experiment, DNA and microscopy samples were collected anaerobically in the anaerobic hood.

In all three experiments we tested for methane-dependent current in different stages (Fig. 1): the gas inflow of $CH_4/CO_2$ 95%/5% was stopped and replaced by Argon/$CO_2$ 95%/5%. For experiment 1, polarisation curves and cyclic voltammetry scans were recorded (Fig. 1). In addition, the gas flow into and out of the BESs was interrupted and 20 mL $^{13}CH_4$ was added to the anolyte while $^{13}CO_2$ and total $CH_4$ were measured by gas chromatography coupled to mass spectrometry using an Agilent 5975 (inert MSD, Agilent, US) and gas chromatography coupled to flame ionization detection with a Hewlett Packard 5890 (Palo Alto, US), respectively. The three experiments were performed over a period of several months to evaluate the reproducibility of the outcome. Negative controls were performed in previous experiments with dead and without biomass[33].

In experiment 4, a mixture of bacterial antibiotics was added to examine the involvement of bacteria in the current production. This mixture consisted of vancomycin HCl (Merck Life Science, Amsterdam, NL), streptomycin sulfate, ampicillin sodium salt, and kanamycin sulfate (VWR, Amsterdam, NL) added at a concentration of 50 µg $mL^{-1}$ per antibiotic. We tested for methane-dependent current before and after the addition of the antibiotics mixture. Additionally, we added 20 mM 2-bromoethanesulfonate (Merck Life Science, Amsterdam, NL) to two of the biological replicates and 50 µg $mL^{-1}$ puromycin dihydrochloride (Merck Life Science, Amsterdam, NL) to two other biological replicates.

### Nucleic acid extraction from 'Ca. Methanoperedens' enrichment culture

RNA samples were taken from experiment 1 and DNA samples were taken from experiment 1 and 2. DNA was isolated following the Powersoil DNeasy kit protocol with the addition of a 10 min bead beating step at 50 $s^{-1}$ (Qiagen, Hilden, Germany). RNA was isolated following the Ribopure Bacteria kit protocol (Thermo Fisher Scientific, Waltham, US), with the addition of a 15 min bead beating step at 50 $s^{-1}$. rRNA depletion was tested according to Phelps et al. 2020[53] but this method was unsuccessful. Instead, the NEB microbe rRNA depletion protocol was used (Macrogen, Seoul, South Korea). The metatranscriptomic datasets were constructed from biological and technical replicates, investigating five different conditions: 0 mV (2 biological, 1 technical replicate, $n = 2$), 200 mV (3 biological replicates, $n = 3$), 400 mV (3 biological replicates, 1 technical replicate, $n = 3$), 600 mV (3 biological replicates, 1 technical replicate, $n = 3$), nitrate condition (3 biological replicates, $n = 3$).

### Metagenomic and metatranscriptomic dataset generation

For metagenome sequencing, the library was prepared using the TruSeq DNA PCR-Free Kit (Illumina, San Diego, CA, United States) and sequenced using Illumina NovaSeq 6000, obtaining 100 M 300 bp paired-end reads (Macrogen, Seoul, South Korea). The quality of reads was determined with FastQC 0.11.9 (Babraham Bioinformatics, Babraham Institute, Cambridge, UK), reads trimmed with BBDuk 37.76 from BBTools (DOE Joint Genome Institute, Walnut Creek, CA, USA) with a minimum read length of 75 bp and co-assembled using MEGAHIT v1.1.1-2-g02102e1[54]. Filtered reads were mapped to contigs with BBmap 37.76 (https://sourceforge.net/projects/bbmap/) and sorted using SAMtools 1.19[55]. For binning, MaxBin 2.0[56], MetaBAT 2.14[57], CONCOCT 1.1.0[58] and BinSanity 0.4.4[59] were used, and consensus bins were constructed with DAS tool 1.1.2[60]. Gene calling and annotation was performed using prodigal[61] and prokka[62]. Relative abundance of specific microorganisms in the microbial community was assessed with Kraken2 2.1.2[63] and Kaiju 1.7.2[64]. For the metatranscriptome sequencing, the library was prepared using the TruSeq Stranded Total RNA Kit (Illumina, San Diego, CA, United States) and sequenced using the Illumina NovaSeq 6000 obtaining 100 M 150 bp paired-end reads. Sequence trimming was performed using Sickle 1.33 and the obtained

reads were pseudo-aligned against the predicted genes of the metagenome using Kallisto v0.46.1, the outcome was represented as transcript per million (tpm) value and counts. The data was further processed by removing genes with less than one count per million and by normalizing the counts using a weighted trimmed mean of the log expression ratios (trimmed mean of M values (TMM)) removing the effect of biological differences between samples making it possible to look at differential expression of genes[65,66]. This method assumes that the majority of the genes is not differentially expressed[66]. All sequencing data are available in the European Nucleotide Archive under BioProject PRJNA995526.

**Visualisation of the biofilm**
Gold electrodes were removed from the BES systems under anoxic conditions. Electrode biofilm was fixed for 1 h anoxically using 4% paraformaldehyde, 100 mM MOPS pH 7.3 after which samples were stored in the fridge in 0.1% paraformaldehyde and 100 mM MOPS pH 7.3 until further use.

**FISH imaging with confocal laser scanning microscopy.** Paraformaldehyde-fixed biomass was dehydrated incubating the samples for 3 min stepwise increasing the ethanol concentration from 25% ethanol to 100%. Pieces of gold electrode were positioned on microscope slides and samples were air-dried for 30 min. A barrier was created around the samples using Silicone grease. Hybridisation buffer containing 180 µL 5 M NaCl, 20 µL 1 M Tris/HCl pH 8.0, 1 µL 10% w/v SDS, 350 µL formamide, 450 µL miliQ water, was carefully added to the samples covering the gold foil in solution while keeping the gold foil attached to the slide. Fluorescence in situ hybridization (FISH) was performed using probes labelled with Cy3, Cy5 and FLUOS that were added to the hybridization buffer at 10% of the total volume: 5'-GGTCCCAAGCCTACCAGT-3'-FLUOS (targeting 'Ca. Methanoperedens')[17], 5'-GTGCTCCCCGCCAATTCCT-3'-Cy3 (targeting archaea)[67], a mixture of three probes 5'-GCTGCCTCCCGTAGGAGT-3'-Cy5[68], 5'-GCAGCCACCCGAGGTGT-3'-Cy5[69], 5'-GCTGCCACCCG-TAGGTGT-3'-Cy5[69], 5'-CCGCAACACCTAGTACTCATC-3'-Cy3 (targeting bacteria)[70]. Probe concentrations were 5 pmol µL$^{-1}$ for Cy3 and Cy5, 8.3 pmol µL$^{-1}$. Slides were incubated at 46 °C for 1.5 h in humidity chambers equilibrated with hybridization buffer. Samples were washed using washing buffer containing per 50 mL 700 µL 5 M NaCl, 500 µL EDTA, 1 mL 1 M Tris/HCl pH 8.0, by adding and removing washing buffer (at 48 °C) several times after which the samples were incubated for 10 min with washing buffer by letting the samples float on top of a styrofoam Eppendorf holder in a 48 °C water bath. The slides were held on ice and washed with ice cold distilled water in the same way as for the washing buffer and subsequently dried using compressed air. Samples were covered with "High Precision" No. 1.5H borosilicate coverslips (Marienfeld-Superior, Lauda-Königshofen, Germany) and visualized using the Leica SP8X confocal point scanning microscope equipped with hybrid HyD detectors and pulsed whitelight laser. Observations were based on five biological replicates ($n = 5$), two from experiment 3 and three from experiment 1.

**Scanning electron microscopy.** Pieces of the anoxically paraformaldehyde-fixed gold mesh film were subjected to an additional fixation with 2% paraformaldehyde and 2.5% glutaraldehyde in 0.1 M PHEM (60 mM Pipes, 25 mM Hepes, 10 mM EGTA, 2 mM MgCl) buffer pH 6.9 for 1.5 h on ice followed by three washes with PHEM Buffer. Samples were contrasted using 1% OsO$_4$ with 1.5% K$_3$[Fe(CN$_6$)] for 1 h on ice followed by four washes with ultrapure water. Dehydration was carried out using a graded acetone series followed by two washes with 100% ethanol and stepwise infiltration with hexamethyldisilazane (HMDS)[71]. Excess HMDS was removed by draining on filter paper. The gold mesh films were air dried overnight before mounting on specimen stubs using carbon tape. Samples were sputter-coated with palladium-gold prior to imaging using a JEOL 6330 Field emission SEM operating at 13 kV using the in-lens detector.

**Transmission electron microscopy.** Samples were processed as described above with the exception that after the dehydration the samples were stepwise infiltrated with EPON resin (2 h 2:1, overnight 1:1, 3 h 1:2, 3 h 1:3 acetone:EPON, overnight 100% EPON) followed by embedding in freshly prepared EPON resin. After curing for 48 h at 60 °C the resin blocks were trimmed by hand using a razor blade. Ultrathin sections (50 nm) sectioned on a diamond knife (Diatome ultra 45) using a Reichert-Jung ultracut E ultramicrotome and mounted on copper grids (100# hexagonal) with a carbon coated formvar film and post-stained using 2% uranyl acetate and Reynolds lead citrate before imaging in a JEOL 1400-Flash TEM operating at 120 kV.

**Reporting summary**
Further information on research design is available in the Nature Portfolio Reporting Summary linked to this article.

## Data availability
All DNA and RNA sequencing data have been deposited in the European Nucleotide Archive under BioProject PRJNA995526. The processed DNA and RNA sequencing data are provided in the Supplemental Data file in this article. The Genome Taxonomy Database used for classification of metagenome-assembled genomes is available under https://gtdb.ecogenomic.org/. The generated bioelectrochemistry and physiology raw data are provided in the Supplementary Information and the Source Data file. Source data are provided with this paper.

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

## Acknowledgements
This study was supported by the SIAM Gravitation grant to MJ funded by NWO [Grant number 024.002.002] and an NWO-VIDI Talent grant to CUW [Grant number VI.Vidi.223.012]. MJ was furthermore supported by the ERC Synergy Grant MARIX [Grant number 854088].

## Author contributions
HTO, RM, TS, MSMJ, AH, TB, and CUW designed the experiments, HTO performed the experiments, MW performed the experiment using anti-biotics, RM prepared the SEM and TEM samples, analysed them using the electron microscope and interpreted the data together with HTO and CUW, JP analysed the FISH samples using the confocal microscope together with HTO and interpreted the data together with HTO and CUW. HTO, MSMJ and CUW wrote and revised the manuscript with contributions from all authors.

## Competing interests
The authors declare no competing interests.
