## [Peer Review File · Nature Communications]

Mechanisms of extracellular electron transfer in anaerobic methanotrophic archaeaReviewer #1 (Remarks to the Author):

The paper by Ouboter et al. presents the initial functional analysis of ANME-2d grown using electrodes as exclusive electron acceptors.

Most notable are the observations of highly expressed genome islands containing multiheme cytochromes and S-layer proteins under electrode-growth conditions but not under nitrate-growth conditions.

Given the limited understanding of ANME mechanisms for extracellular electron transfer (EET), this investigation stands as the first comprehensive exploration of EET in an ANME-archaea, where autonomous growth at the anode surface occurs independently of partners.

To enhance the manuscript, the authors may want to consider the following points.

Major points:

- The involvement of OMC nanowires cannot be conclusively inferred from the current data. While the results hold significance, it is prudent not to overstate the role of OMC-based nanowires without additional evidence. The authors could propose this connection, but should avoid presenting it as fact without further substantiation.
- Review references for accurate alignment with their corresponding sentences. Proper referencing will enhance the credibility of the content.
- Elaborate on the schematic representation of experimental procedures (possibly in the figure legend), including details on replication for RNA work and MAGs, to provide a comprehensive understanding of the methodology.
- The bioelectrochemical results warrant a more thorough discussion. Provide insights into the rationale behind performing polarization curves and the anticipated versus observed outcomes. Delve into the interpretation of redox peaks in cyclic voltammograms (CVs) and their potential implications, aligning these findings with the research hypothesis.
- Note that transmission electron microscopy (TEM) cannot determine cell viability. Given the diverse impacts of sample preparation on archaea and bacterial cells, refrain from making definitive statements about cellular vitality unless supported by additional information. Remove any assertions about cell viability that lack corroborating data.
-
- Clearly indicate in both text and figures that the presented model depicts a proposed extracellular electron transfer metabolism in *Ca. Methanoperedens*. It is not a definitively well-characterized EET metabolism.

Minor comments:

Line 31-32: Please remove or rephrase the reference to nanowire involvement. Your results currently demonstrate the presence of related genes and transcripts but not the actual presence of nanowires on cell surfaces. Further evidence is needed to establish the connection.

Line 50: Does reference 28 really pertain to iron oxides in sediments?

Line 70: Throughout the text, please emphasize that comparisons are made with a no-electrode condition, where cells are cultivated using nitrate as the terminal electron acceptor.

Fig. 4: Panels A, E, I seem to offer limited information. Consider relocating them to the supplementary materials to allow the remaining panels to be enlarged for better visibility of cells.

Line 173: - Remove "live"

Line 175: - Edit to eliminate "appeared to be empty and therefore dead."

Line 239: What is the homology to omcZ? Generally, the homology of OMCs between *Geobacter* species is weak. How does the "omcZ" of *Methanoperedens* compare to other omcZ-homologues in *Geobacter* species?

Line 273: The current results cannot lead to an overview of the actual metabolism. Please rephrase to specify that this is a proposed model of EET metabolism for *Methanoperedens*.

Lines 309-314: Clarify the term "fused." Does your MAG data provide structural insights confirming the fusion of MHC with the S-layer? It does not, so adjust the text accordingly.

Line 342: Revise sentences regarding dead/live cells. Alternatively, consider conducting live-dead staining experiments.

Line 430: Were cells fixed with PFA before FISH?

Reviewer #2 (Remarks to the Author):

Ouboter et al present experiments conducted on bioelectrochemical systems dominated by "*Candidatus methanoperedens*" an anaerobic methanotroph. The authors clearly demonstrate that after biofilm formation the current is dependent, to varying degrees, on the continuous addition of methane, as well as the activity of MCR by use of MCR-specific inhibitors. Analysis of transcriptomic data helps shed light on possible routes for electrons to exit "*Ca. methanoperedens*". These results are largely consistent with existing studies, and the additional experimental conditions and analysis presented here make this a useful contribution to the field that is worthy of publication.

The only major concern with the current manuscript is an over-interpretation of the data to support the notion that "*Ca. methanoperedens*" can carry this process out all on its own. e.g.:

Line 32: "Our findings furthermore indicate that bioelectrochemical cells might be powerful tools for the cultivation, and possibly isolation, of uncultured electroactive microorganisms."

Line 346: "We present evidence for direct current production and extracellular electron transfer by "*Ca. methanoperedens*" during methane oxidation without the need of syntrophic partner microorganisms".

Unfortunately the data supporting this claim is far too weak and any reference to it should be removed from the abstract and discussion prior to acceptance of this manuscript.

This is a mixed culture, and it is abundantly clear from the microscopy that there is strong colonization of the electrode by *Geobacter* (Fig 4L). There is massive current generation between days 0-5 of BES start-up which the authors do not address, but this probably corresponds to colonization and robust growth of a *Geobacter* biofilm. This is not a trivial amount of current, in fact, it appears that the vast majority of electrons transferred to the electrode in the experiments occurs in the first 10 days or so, which is well before methane-dependent current is demonstrated.

All the microscopy analysis shows that "*Ca. methanoperedens*" is layered on top of this presumably conductive *Geobacter* biofilm. While some of the TEM images show lysed bacterial cells near the electrode, the FISH images would seem to contradict massive lysis and loss of cytoplasmic components in the *Geobacter* cells colonizing the electrodes. It seems incredibly likely that these BES are initially colonized by *Geobacter*, which must be growing on something that

comes along in the inoculum, and after all of that initial growth is completed then there is slow current generation by methane oxidation. The microscopy all shows there is no direct colonization or contact between the electrode and "Ca. methanoperedens". The antibiotic experiments do not help rule out the importance of bacteria in the formation of this BES system because they were added well after the community had developed.

To demonstrate that "Ca. methanoperedens" can do this on its own without bacteria the authors would need to add antibiotics upon inoculation of the BES system. If this inhibited growth of a Geobacter biofilm and resulted in colonization of the electrode by "Ca. methanoperedens" then the authors could make these claims. Unfortunately, this simple experiment was not done, so it seems just as likely that "Ca. methanoperedens" requires a conductive Geobacter biofilms to conduct electron for it, or that some sort of intermediate carbon compound such as acetate is produced by "Ca. methanoperedens" and taken up by Geobacter. Although the authors point out in line 339 that Geobacter is sensitive to streptomycin and kanamycin, the inhibition of growth on plates discussed in ref 55 does not mean that these antibiotics in this system will inhibit catabolic activity. Slow growing biofilms are famously difficult to treat with antibiotics.

Minor comments:

Line 31: an -> a

Line 85: description of exp 3 seems incomplete "the microbial community was incubated for 9 weeks at 0mV vs SHE in a BES." Figure shows that there is famine (-400mV) for 3 of those weeks though?

Fig 1: spelling error: "polarisation" scan

Line 100: It is not clear how the data in fig 2B could result in average value of 39+/-8.8mV. The max current is at 0mV looks to be 39 or less, with the other three potentials below 30. Check math?

Fig 2C: X-axis mislabeled

Fig S4: Explain what the symbols represent. Replicate chambers?

Lines 134-143: Since this starvation experiment is not returned to in the Discussion, I would encourage the authors to make a clearer statement about their interpretation of the result, even if the result is inconclusive. The justification is to see if "Ca. methanoperedens" can be reversed to make methane at low applied potentials. What are we to take from the small amount of methane production? The reversibility of ANME is an interesting question that comes up again and again in the literature, so this is an interesting experiment. But perhaps without a condition where the electrode was detached to prove that the methane production was dependent on the current from the electrode, it would be difficult to claim cathodic methanogenesis from this experiment.

Line 296: chemolithotrophic autotroph -> chemolithoautotroph

The mentioned line numbers refer to the tracked change version of the revised manuscript.

Reviewer #1 (Remarks to the Author):

The paper by Ouboter et al. presents the initial functional analysis of ANME-2d grown using electrodes as exclusive electron acceptors.

Most notable are the observations of highly expressed genome islands containing multiheme cytochromes and S-layer proteins under electrode-growth conditions but not under nitrate-growth conditions.

Given the limited understanding of ANME mechanisms for extracellular electron transfer (EET), this investigation stands as the first comprehensive exploration of EET in an ANME-archaea, where autonomous growth at the anode surface occurs independently of partners.

We thank the reviewer for their appreciation of our manuscript

To enhance the manuscript, the authors may want to consider the following points.

Major points:

- The involvement of OMC nanowires cannot be conclusively inferred from the current data. While the results hold significance, it is prudent not to overstate the role of OMC-based nanowires without additional evidence. The authors could propose this connection, but should avoid presenting it as fact without further substantiation.

Thank you for the comment. We have checked the manuscript carefully and have tuned down the statements on the connection between the two upregulated gene clusters and EET (e.g. line 30-33; line 227-278)

- Review references for accurate alignment with their corresponding sentences. Proper referencing will enhance the credibility of the content.

Thank you – we have gone through the manuscript carefully and amended this at several places (e.g. line 49, line 59-60, line 326, line 328, line 332).

- Elaborate on the schematic representation of experimental procedures (possibly in the figure legend), including details on replication for RNA work and MAGs, to provide a comprehensive understanding of the methodology.

We have included this information now in the Figure legend (line 92-99), in addition to mentioning it in the Materials part (line 441-444).

- The bioelectrochemical results warrant a more thorough discussion. Provide insights into the rationale behind performing polarization curves and the anticipated versus observed outcomes. Delve into the interpretation of redox peaks in cyclic voltammograms (CVs) and their potential implications, aligning these findings with the research hypothesis.

We thank the referee for this suggestion. We added more context to the polarization curves in the Results section (line 120-123), stating that the observed intercept with the x-axis was in line with methane and not acetate as a proxy for organic material as electron donor. We also expanded the discussion of the cyclic voltammograms (line 113-117), including naming of the two observed redox centers at -180 mV and +100 mV. The implications of these findings were further discussed (line 352-355).

- Note that transmission electron microscopy (TEM) cannot determine cell viability. Given the diverse impacts of sample preparation on archaea and bacterial cells, refrain from making definitive statements about cellular vitality unless supported by additional information. Remove any assertions about cell viability that lack corroborating data.

We have removed mentioning of cell viability as a result of TEM imaging (line 186-188, line 196, line 376).

-

- Clearly indicate in both text and figures that the presented model depicts a proposed extracellular electron transfer metabolism in *Ca. Methanoperedens*. It is not a definitively well-characterized EET metabolism.

We indicated that it is a proposed mechanism (line 277-278, line 297).

Minor comments:

Line 31-32: Please remove or rephrase the reference to nanowire involvement. Your results currently demonstrate the presence of related genes and transcripts but not the actual presence of nanowires on cell surfaces. Further evidence is needed to establish the connection.

The sentence has been re-phrased according to the referee's suggestion (line 30-32).

Line 50: Does reference 28 really pertain to iron oxides in sediments?

The citation was moved to earlier in the sentence to reflect the content of the respective review article (line 49).

Line 70: Throughout the text, please emphasize that comparisons are made with a no-electrode condition, where cells are cultivated using nitrate as the terminal electron acceptor.

This was added to the sentence (line 69, line 285-286).

Fig. 4: Panels A, E, I seem to offer limited information. Consider relocating them to the supplementary materials to allow the remaining panels to be enlarged for better visibility of cells.

We agree with the referee and have removed panels A, E, and I from Figure 4.

Line 173: - Remove "live"

Removed (line 194).

Line 175: - Edit to eliminate "appeared to be empty and therefore dead."

Removed (line 196).

Line 239: What is the homology to *omcZ*? Generally, the homology of OMCs between *Geobacter* species is weak. How does the "*omcZ*" of *Methanoperedens* compare to other *omcZ*-homologues in *Geobacter* species?

With electron conduit c-type cytochromes the structural homology is more important than the sequence identity, as the referee implies, therefore we have added a sentence to refer to the structural homology observed by Guo *et al.* who compared the *Ca. Methanoperedens* AlphaFold model to their observed *Geobacter* CryoEM model (line 265-267)

Line 273: The current results cannot lead to an overview of the actual metabolism. Please rephrase to specify that this is a proposed model of EET metabolism for *Methanoperedens*.

Changed as requested (line 299).

Lines 309-314: Clarify the term "fused." Does your MAG data provide structural insights confirming the fusion of MHC with the S-layer? It does not, so adjust the text accordingly.

We changed the text to make it clear that we analyzed the gene and not the protein (line 282-285).

Line 342: Revise sentences regarding dead/live cells. Alternatively, consider conducting live-dead staining experiments.

We have changed the text to reflect our experimental observations, removing the mention of live/dead cells (line 186-188, line 194, line 196, line 376).

Line 430: Were cells fixed with PFA before FISH?

Yes, they were, and this is mentioned in line 464 before the start of the FISH paragraph because all microscopy samples were fixed with PFA, not only the FISH samples. For clarification, we have added this information to the first sentence of the FISH imaging paragraph (line 467) as well as the SEM sample preparation paragraph (line 490).

Reviewer #2 (Remarks to the Author):

Ouboter et al present experiments conducted on bioelectrochemical systems dominated by "*Candidatus methanoperedens*" an anaerobic methanotroph. The authors clearly demonstrate that after biofilm formation the current is dependent, to varying degrees, on the continuous addition of methane, as well as the activity of MCR by use of MCR-specific inhibitors. Analysis of transcriptomic data helps shed light on possible routes for electrons to exit "*Ca. methanoperedens*". These results are largely consistent with existing studies, and the additional experimental conditions and analysis presented here make this a useful contribution to the field that is worthy of publication.

We thank the reviewer for their appreciation of our work.

The only major concern with the current manuscript is an over-interpretation of the data to support the notion that "*Ca. methanoperedens*" can carry this process out all on its own. e.g.:

Line 32: "Our findings furthermore indicate that bioelectrochemical cells might be powerful tools for the cultivation, and possibly isolation, of uncultured electroactive microorganisms."

We've removed the corresponding line from the manuscript (line 33-34).

Line 346: "We present evidence for direct current production and extracellular electron transfer by "Ca. methanoperedens" during methane oxidation without the need of syntrophic partner microorganisms".

We have rephrased this statement: "we present evidence for current production and extracellular electron transfer by 'Ca. Methanoperedens' during methane oxidation, either through DIET with Geobacter sp. or independently" (line 383-384).

Unfortunately the data supporting this claim is far too weak and any reference to it should be removed from the abstract and discussion prior to acceptance of this manuscript.

Done (line 31-33, line 371-375, line 383-384).

This is a mixed culture, and it is abundantly clear from the microscopy that there is strong colonization of the electrode by Geobacter (Fig 4L). There is massive current generation between days 0-5 of BES start-up which the authors do not address, but this probably corresponds to colonization and robust growth of a Geobacter biofilm. This is not a trivial amount of current, in fact, it appears that the vast majority of electrons transferred to the electrode in the experiments occurs in the first 10 days or so, which is well before methane-dependent current is demonstrated.

We mention in the manuscript that during the start-up phase, most electricity comes from processes other than methanotrophy (Figure 3 and associated results/discussion). While it is interesting to further study this initial peak, we think that it will distract the reader from our main experiments and results that are drawn from later stages of the experiment if we focus on it further. As the referee agrees, we observe colonization of the electrode by Geobacter which we mention at several places in the manuscript so the essential information is contained.

All the microscopy analysis shows that "Ca. methanoperedens" is layered on top of this presumably conductive Geobacter biofilm. While some of the TEM images show lysed bacterial cells near the electrode, the FISH images would seem to contradict massive lysis and loss of cytoplasmic components in the Geobacter cells colonizing the electrodes. It seems incredibly likely that these BES are initially colonized by Geobacter, which must be growing on something that comes along in the inoculum, and after all of that initial growth is completed then there is slow current generation by methane oxidation. The microscopy all shows there is no direct colonization or contact between the electrode and "Ca. methanoperedens". The antibiotic experiments do not help rule out the importance of bacteria in the formation of this BES system because they were added well after the community had developed.

While we agree to some extent with the referee that more data are needed to unequivocally consolidate our findings, we do think that our data strongly indicate that Ca. Methanoperedens is the driver of current production, as the addition of the methanotroph inhibitor (BES) almost completely abolished current production. In the unlikely case Geobacter would have been responsible for (the majority of) current production, we should have observed - rather than a sudden decrease of current - an increase of current as the decaying biomass of Ca. Methanoperedens would have supplied Geobacter with ample

electron donor to produce electricity (and the polarization curves would have shown a more negative x axis intercept). It is still possible that (some) electricity was produced by *Geobacter* through DIET with *Ca. Methanoperedens*, which is mentioned explicitly in the text. To give justice to the concerns raised by the reviewer we have toned down our conclusions to transparently reflect what we found in the experiments (line 31-33, line 371-375, line 383-384).

We have removed the statement about the dead bacterial cells in the TEM micrographs (line 186-188, line 194, line 196, line 376).

To demonstrate that “*Ca. methanoperedens*” can do this on its own without bacteria the authors would need to add antibiotics upon inoculation of the BES system. If this inhibited growth of a *Geobacter* biofilm and resulted in colonization of the electrode by “*Ca. methanoperedens*” then the authors could make these claims. Unfortunately, this simple experiment was not done, so it seems just as likely that “*Ca. methanoperedens*” requires a conductive *Geobacter* biofilms to conduct electron for it, or that some sort of intermediate carbon compound such as acetate is produced by “*Ca. methanoperedens*” and taken up by *Geobacter*. Although the authors point out in line 339 that *Geobacter* is sensitive to streptomycin and kanamycin, the inhibition of growth on plates discussed in ref 55 does not mean that these antibiotics in this system will inhibit catabolic activity. Slow growing biofilms are famously difficult to treat with antibiotics.

We thank the referee for this suggestion. We agree that it would be an interesting experiment and follow-up work will indeed focus in this direction. Given the time-intensive nature of conducting the said experiments with such slow growing and delicate cultures as *Ca. Methanoperedens*, this is by no means a ‘simple and fast’ experiment (it would be if we would culture *Geobacter*!). Considering the wealth of data we already provide, we think that this is beyond the scope of this manuscript.

Minor comments:

Line 31: an -> a

Changed (line 31).

Line 85: description of exp 3 seems incomplete “the microbial community was incubated for 9 weeks at 0mV vs SHE in a BES.” Figure shows that there is famine (-400mV) for 3 of those weeks though?

This information was added to the text (line 86-87).

Fig 1: spelling error: “polarisation” scan

Changed.

Line 100: It is not clear how the data in fig 2B could result in average value of 39 ± 8.8 mV. The max current is at 0mV looks to be 39 or less, with the other three potentials below 30. Check math?

Indeed, we are sorry for this oversight; it's 30 ± 8.8 mA m⁻² and was changed in the text accordingly.

Fig 2C: X-axis mislabeled

Changed.

Fig S4: Explain what the symbols represent. Replicate chambers?

We thank the referee for spotting this omission, it indeed represents biological replicates. We have added this information to the figure legend.

Lines 134-143: Since this starvation experiment is not returned to in the Discussion, I would encourage the authors to make a clearer statement about their interpretation of the result, even if the result is inconclusive. The justification is to see if "Ca. methanoperedens" can be reversed to make methane at low applied potentials. What are we to take from the small amount of methane production? The reversibility of ANME is an interesting question that comes up again and again in the literature, so this is an interesting experiment. But perhaps without a condition where the electrode was detached to prove that the methane production was dependent on the current from the electrode, it would be difficult to claim cathodic methanogenesis from this experiment.

We added a statement in the results section indicating that possibly Ca. Methanoperedens can reverse their metabolism at extremely low metabolic rates that are probably insignificant in the natural habitat (line 155-160).

Line 296: chemolithotrophic autotroph -> chemolithoautotrophy

Changed (line 326).